# Neural Deconstruction Search for Vehicle Routing Problems

## Abstract

Autoregressive construction approaches generate solutions to vehicle routing problems in a step-by-step fashion, leading to high-quality solutions that are nearing the performance achieved by handcrafted, operations research techniques. In this work, we challenge the conventional paradigm of sequential solution construction and introduce an iterative search framework where solutions are instead *deconstructed* by a neural policy. Throughout the search, the neural policy collaborates with a simple greedy insertion algorithm to rebuild the deconstructed solutions. Our approach surpasses the performance of state-of-the-art operations research methods across three challenging vehicle routing problems of various problem sizes.

## 1 Introduction

Methods that can learn to solve complex optimization problems have the potential to transform decision-making processes across virtually all domains. It is therefore unsurprising that learning-based optimization approaches have garnered significant attention and yielded substantial advancements (Bello et al., 2016; Kool et al., 2019; Kwon et al., 2020). Notably, reinforcement learning (RL) approaches are particularly promising because they do not rely on a pre-defined training set of representative solutions and can develop new strategies from scratch for novel optimization problems. These methods generally construct solutions *incrementally* through a sequential decision-making process and have been successfully applied to various vehicle routing problems.

Despite recent progress, learning-based methods for combinatorial optimization (CO) problems usually fall short of outperforming the state-of-the-art techniques from the operations research (OR) community. For instance, while some new construction approaches for the capacitated vehicle routing problem (CVRP) have surpassed the LKH3 solver (Helsgaun, 2000), they still struggle to match the performance of the state-of-the-art HGS solver (Vidal et al., 2012), particularly for larger instances with over 100 nodes. One reason for this is their inability to explore as many solutions as traditional approaches within the same amount of time. Given the limitations of current construction approaches, we propose challenging the traditional paradigm of sequential solution construction by introducing a novel iterative search framework, *neural deconstruction search (NDS)*, which instead deconstructs solutions using a neural policy.

NDS is an iterative search method designed to enhance a given solution through a two-phase process involving deconstruction and reconstruction along the lines of large neighborhood search (LNS) (Shaw, 1998) and ruin-and-recreate (Schrimpf et al., 2000) paradigms. The deconstruction phase employs a deep neural network (DNN) to determine the customers to be removed from the tours of the current solution. This is achieved through a sequential decision-making process, in which nodes are removed one at a time based on the network's guidance. The reconstruction phase utilizes a straightforward greedy insertion algorithm, which inserts customers in the order given by the neural network at the locally optimal positions. The core concept of NDS is shown in Figure 1. Note that NDS is trained using reinforcement learning, which makes it adaptable to problems for which no reference solutions are available for training.

The overall concept of modifying a solution by first removing some solution components and then conducting a rebuilding step has been successfully used in various vehicle routing problem methods. Non-learning based methods that use this concept include the rip-up and reroute method from Dees & Smith (1981), LNS from Shaw (1998), and the ruin and recreate method from Schrimpf et al.

Figure 1: Improving a solution via neural deconstruction.

(2000). Learning-based methods have also harnessed this paradigm. The local rewriting method from Chen & Tian (2019), neural large neighborhood search from Hottung & Tierney (2020), and the random reconstruction technique introduced in Luo et al. (2023) employ a DNN during the reconstruction phase. The approaches from Li et al. (2021) and Falkner & Schmidt-Thieme (2023) both generate different subproblems for a given solution and then use a DNN to choose which subproblem should be considered in the reconstruction phase.

NDS has been designed with the goal of achieving a fast search procedure without sacrificing the high-quality search guidance of a DNN. For medium-sized CVRP instances with 500 customers, state-of-the-art OR approaches such as SISRs (Christiaens & Vanden Berghe, 2020) can examine upwards of 270k solutions per second, however neural combinatorial optimization approaches, like POMO (Kwon et al., 2020), can only observe around 10k per second. In contrast, NDS can process 120k solutions per second, significantly more than existing neural construction techniques. When combined with a powerful deconstruction DNN, NDS is able to outperform state-of-the-art OR approaches like SISRs and HGS in similar wall-clock time.

We evaluate NDS on several challenging problems, including the CVRP, the vehicle routing problem with time windows (VRPTW), and the price-collecting vehicle routing problem (PCVRP). NDS demonstrates substantial performance gains compared to existing learned construction methods and surpasses state-of-the-art OR methods across various routing problems of different sizes. To the best of our knowledge, NDS is the first learning-based approach that achieves this milestone.

In summary, we provide the following contributions:

- We propose to use a learned deconstruction policy in combination with a simple greedy insertion algorithm.
- We introduce a novel training procedure designed to learn effective deconstruction policies.
- We present a new network architecture optimized for encoding the current solution.
- We develop a high-performance search algorithm specifically designed to leverage the parallel computing capabilities of GPUs.

## 2 LITERATURE REVIEW

**Construction Methods** The introduction of the pointer network architecture by Vinyals et al. (2015) marked the first autoregressive, deep learning-based approach for solving routing problems. In their initial work, the authors employ supervised learning to train the models, demonstrating its application to the traveling salesperson problem (TSP) with 50 nodes. Building on this, Bello et al. (2016) propose using reinforcement learning to train pointer networks, showcasing its effectiveness in addressing larger TSP instances.

For the more complex CVRP, the first learning-based construction methods were introduced by Nazari et al. (2018) and Kool et al. (2019). Recognizing the symmetries inherent in many combinatorial optimization problems, Kwon et al. (2020) develop POMO, a method that leverages these symmetries to improve exploration of the solution space during both training and testing. Extending this concept, Kim et al. (2022) propose a general-purpose symmetric learning framework.

Various techniques have been proposed to enhance performance in neural combinatorial optimization. For instance, Hottung et al. (2022) introduce efficient active search, which updates a subset of parameters during inference. Choo et al. (2022) propose SGBS, combining Monte Carlo tree search

with beam search to guide the search process more effectively. Additionally, Drakulic et al. (2023) and Luo et al. (2023) focus on improving out-of-distribution generalization by re-encoding the remaining subproblem after each construction step. To enhance solution diversity during sampling, Grinsztajn et al. (2022) and Hottung et al. (2024) explore approaches that learn a set of policies, rather than a single policy.

Instead of constructing solutions autoregressively, some approaches predict heat maps that highlight promising solution components (e.g., arcs in a graph), which are then used in post-hoc searches to construct solutions (Joshi et al., 2019; Fu et al., 2021; Kool et al., 2022b; Min et al., 2023). Other approaches focus on more complex variants of routing problems, such as the VRPTW (Falkner & Schmidt-Thieme, 2020; Kool et al., 2022a; Berto et al., 2024a;b).

**Improvement Methods**   Improvement methods focus on iteratively refining a given starting solution. In addition to the ruin-and-recreate approaches discussed in the introduction, several other methods aim to enhance solution quality through iterative adjustments. For instance, Ma et al. (2021) propose learning to iteratively improve solutions by performing local modifications. Similarly, several works have guided the $k$-opt heuristic for vehicle routing problems (Wu et al., 2019; da Costa et al., 2020), although they are constrained by a fixed, small $k$. More recently, Ma et al. (2023) introduced a method capable of handling any $k$. Furthermore, Ye et al. (2024a) and Kim et al. (2024) integrate learning-based approaches with ant colony optimization to allow for a more extensive search phase. Additionally, several divide-and-conquer methods have been developed to address large-scale routing problems (Kim et al., 2021; Li et al., 2021; Ye et al., 2024b).

## 3   NEURAL DECONSTRUCTION SEARCH

### 3.1   DECONSTRUCTION POLICY

For solution deconstruction, a neural policy is employed to sequentially select customers for removal from a given solution. We model this selection process as a Markov decision process. Let $s$ be a feasible solution to a vehicle routing problem (VRP) instance $l$, which involves customers $c_1, \ldots, c_N$. A policy network $\pi_\theta$, parameterized by $\theta$, is used to select $M$ customers for removal. At each step $m \in \{1, \ldots, M\}$, an action $a_m \in \{1, \ldots, N\}$ is chosen according to the probability distribution $\pi_\theta(a_m \mid l, s, v, a_{1:m-1})$, where $a_m$ corresponds to selecting customer $c_{a_m}$, $l$ is the instance, $s$ is the solution, $v$ is a random seed, and $a_{1:m-1}$ are the previous actions. We condition the policy on a random seed $v$ to encourage more diverse rollouts as explained in Hottung et al. (2024). Each seed is a randomly generated binary vectors of dimension $d_v$ (we set $d_v = 10$ in all experiments). Finally, after all $M$ customers are selected the reward can be computed as discussed in the following sections.

### 3.2   TRAINING

The deconstruction policy in NDS is trained using reinforcement learning. During the training process, solutions are repeatedly deconstructed and reconstructed, aiming to discover a deconstruction policy that facilitates the reconstruction of high-quality solutions. Algorithm 1 outlines our training procedure. It is important to implement the algorithm in a way that allows processing batches of instances in parallel to ensure efficient training. However, for clarity, the pseudocode presented describes the training process for a single instance at a time.

The main training loop runs until a termination criterion (such as the number of processed instances) is met. In each iteration of the loop, a new instance and its corresponding solution are generated in lines 4-8. The solution is then repeatedly deconstructed and reconstructed for $I$ iterations (lines 9-18), during which gradients are computed based on the rewards obtained. After completing $I$ iterations, the gradients are accumulated, and the network parameters are updated using the learning rate $\alpha$. The following section provides a more detailed explanation of this process.

At the start of each iteration of the training loop, a new instance $l$ and its corresponding solution $s$ are generated. The instance is sampled from the same distribution as the test instances. In line 5, an initial solution is constructed using a simple procedure: for an instance with $N$ customers, we generate $N$ tours, each containing one customer. In lines 6-8, this initial solution is iteratively

---

**Algorithm 1** NDS Training

---

1: **procedure** TRAIN(Iterations per instance $I$, rollouts per solution $K$, improvement steps $J$)
2:     Initialize policy network $\pi_\theta$
3:     **while** Termination criteria not reached **do**
4:         $l \leftarrow$ GENERATEINSTANCE$(\,)$
5:         $s \leftarrow$ GENERATESTARTSOLUTION$(l)$
6:         **for** $j = 1, \ldots, J$ **do**
7:             $s \leftarrow$ IMPROVEMENTSTEP$(s, \pi_\theta)$      ▷ Improve solution using procedure shown in Figure 2
8:         **end for**
9:         **for** $i = 1, \ldots, I$ **do**
10:           $\{\tau_1, \tau_2, \ldots, \tau_K\} \leftarrow$ ROLLOUTPOLICY$(\pi_\theta, l, s, K)$
11:           $\bar{s}_k \leftarrow$ REMOVECUSTOMERS$(s, \tau_k)$    $\forall k \in \{1, \ldots, K\}$
12:           $s'_k \leftarrow$ GREEDYINSERTION$(\bar{s}_k, \tau_k)$    $\forall k \in \{1, \ldots, K\}$
13:           $r_k \leftarrow \max(\text{OBJ}(s) - \text{OBJ}(s'_k), 0)$    $\forall k \in \{1, \ldots, K\}$      ▷ Calculate reward
14:           $b \leftarrow \frac{1}{K} \sum_{k=1}^{K} r_k$      ▷ Calculate baseline
15:           $k^* = \arg\max_{k \in \{1, \ldots, K\}} r_k$
16:           $g_i \leftarrow (r_{k^*} - b) \nabla_\theta \log \pi_\theta(\tau_{k^*} | s, s, v_{k^*})$      ▷ Calculate gradients
17:           $s \leftarrow s'_{k^*}$      ▷ Update $s$ with best found solution
18:         **end for**
19:         $\theta \leftarrow \theta + \alpha \sum_{i=1}^{I} g_i$      ▷ Optimizer step with accumulated gradients
20:     **end while**
21: **end procedure**

---

improved through $J$ improvement steps of the NDS search procedure, which are detailed in Section 3.4. By refining $s$ with NDS's main search component before the training rollouts, we ensure that the training focuses on improving non-trivial solutions.

In lines 9 to 18, the solution $s$ is improved over $I$ iterations. At the start of each iteration, the policy $\pi_\theta$ is used to sample $K$ rollouts $\tau_1, \tau_2, \ldots, \tau_K$, using $K$ different, random seed vectors $v_0, \ldots, v_k$. Each rollout is a sequence of $M$ actions that specifies the indices of customers to be removed from the tours in solution $s$. Each rollout $\tau_k$ is individually applied to deconstruct solution $s$ by removing the specified customers, yielding $K$ deconstructed solutions $\bar{s}_1, \ldots, \bar{s}_K$. These deconstructed solutions are then repaired using the greedy insertion algorithm, which is described in more detail below. Next, the reward $r_k$ is calculated for each rollout $\tau_k$, based on the difference in cost between the original solution $s$ and the reconstructed solution $s'_k$. Importantly, the reward is constrained to be non-negative, encouraging the learning of risk-taking policies. In lines 14 to 16, the gradients are computed using the REINFORCE method. The overall probability of generating a rollout $\tau_k$ is given by $\pi_\theta(\tau_k \mid l, s, v_k) = \prod_{m=1}^{M} \pi_\theta(a_m \mid l, s, v_k, a_{1:m-1})$. The baseline $b$ is set as the average cost of all rollouts. Gradients are only calculated with respect to the best-performing rollout, denoted $k^*$, to encourage diversity in the solutions as proposed by Grinsztajn et al. (2022). Finally, at the end of each iteration, the solution $s$ is replaced by the reconstructed solution with the highest reward.

**Greedy Insertion** The greedy insertion procedure reintegrates the customers removed by the policy, inserting them one by one into either existing or new tours. Specifically, if $M$ customers have been removed, the procedure performs $M$ iterations, where in each iteration, a single customer $c_{a_m}$ is inserted. At each iteration $m$, the cost of inserting customer $c_{a_m}$ at every feasible position in the current tours is evaluated. Throughout this process, various constraints, such as vehicle capacity limits, are taken into account. If at least one feasible insertion point is found within an existing tour, the customer $c_{a_m}$ is placed at the position that incurs the least additional cost. If no feasible insertion is available, a new tour is created for customer $c_{a_m}$.

The order in which removed customers are reinserted significantly impacts the overall performance. We reinsert customers either in the order determined by the neural network or at random. Allowing the network to control the reinsertion order gives it control over the reconstruction process, enabling it to find ordering strategies that lead to better reconstructed solutions. If customers are ordered at random, a deconstructed solution should be reconstructed multiple times using different insertion orders. This can provide more stable learning signal during training.

### 3.3 MODEL ARCHITECTURE

We design a transformer-based architecture that consists of an encoder and a decoder. The encoder is used to generate embeddings for all nodes based on the instance $l$ and the current solution $s$. The decoder is used to decode a sequence of actions based on these embeddings in an iterative fashion.

#### 3.3.1 ENCODER

The encoder processes the input features $x_i$ for each of the $N + 1$ nodes, where $x_0$ corresponds to the depot's features, alongside the current solution $s$ that needs to be encoded. Initially, each input vector $x_i$ is mapped to a 128-dimensional node embedding $h_i$ through a linear transformation. The embeddings $h_0, \ldots, h_N$ are sequentially processed through several layers. First, two attention layers encode static instance information. Next, a message passing layer allows information exchange between consecutive nodes in the solution. This is followed by a tour embedding layer, which computes embeddings for each tour within the solution. Finally, two additional attention layers refine the representations. The attention mechanisms employed are consistent with those used in prior work (e.g., Kwon et al. (2020)), and detailed descriptions are omitted here for brevity.

**Message Passing Layer** The message passing layer updates the embedding of a customer $c_i$ by incorporating information from its immediate neighbors (i.e., nodes that are visited before and after $c_i$ in the solution $s$). Specifically, the embedding $h_i$ of customer $c_i$ is updated as follows:

$$h_i' = \text{Norm}\left(h_i + \text{FF}\left(\text{ReLU}\left(W^3\left[h_i; W^1 h_{\text{prev}(i)} + W^2 h_{\text{next}(i)}\right]\right)\right)\right)$$

In this equation, $\text{prev}(i)$ and $\text{next}(i)$ represent the indices of the nodes immediately preceding and following $c_i$ in the solution $s$. The weight matrices $W^1$ and $W^2$ are used to transform the embeddings of these neighboring nodes, while $W^3$ is applied to the concatenated vector of $h_i$ and the aggregated embeddings from the neighbors. The ReLU activation function introduces non-linearity into the transformation. The output of this transformation is processed through a feed-forward network, which consists of two linear layers with a ReLU activation function in between. The resulting output, combined with the original embedding $h_i$ via a skip connection, is then normalized using instance normalization.

**Tour Encoding Layer** The tour encoding layer updates the embedding of each customer $c_i$ by incorporating information from the tour they are part of. To this end, a tour embedding is first computed using mean aggregation of the embeddings of all customers within the same tour, and this aggregated tour embedding is then used to update the individual customer embeddings. Specifically, the embedding $h_i$ of customer $c_i$ is updated as follows:

$$\hat{h}_i = \text{Norm}\left(h_i' + \text{FF}\left(\text{ReLU}\left(W^4\left[h_i'; \sum_{j \in \mathcal{T}(i)} h_j'\right]\right)\right)\right),$$

where $\mathcal{T}(i)$ denotes the set of customers in the same tour as customer $c_i$ and $W^4$ is a weight matrix. This layer captures important information about which customers belong to the same tour in the current solution, without considering their specific positions within the tour.

#### 3.3.2 DECODER

Given the node embeddings generated by the encoder, the decoder is responsible for sequentially selecting customers for removal. The overall architecture of our decoder is identical to that of Hottung et al. (2024), which utilizes a multi-head attention mechanism (Vaswani et al., 2017) followed by a pointer mechanism (Vinyals et al., 2015). This architecture has been widely used in many routing problems methods (Kool et al., 2019; Kwon et al., 2020).

Our approach differs from previous works in that we account for the already selected customers at each decision step. This contrasts with construction-based methods, where each decision is independent of prior selections. To address this, we integrate a gated recurrent unit (GRU) (Cho, 2014), which is used to compute the query for the multi-head attention mechanism. At each decision step, the GRU takes the embedding of the previously selected customer as input, updating its internal state to incorporate past decisions.

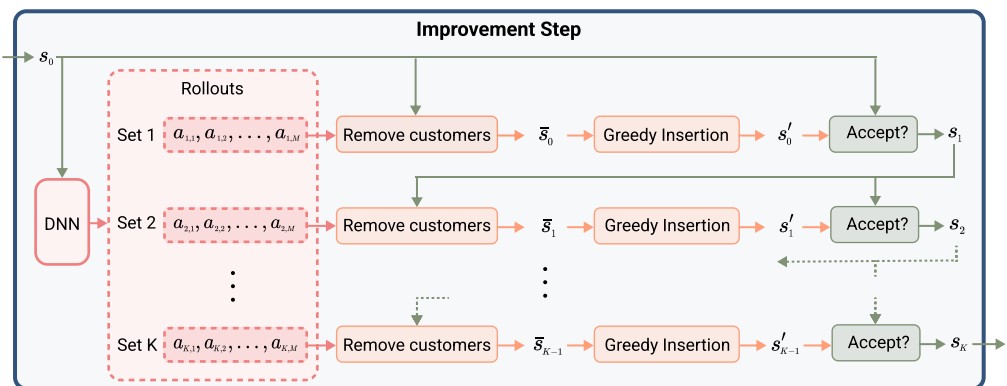

Figure 2: Improvement step of NDS.

## 3.4 SEARCH

At test time, we leverage the learned policy within a search framework that supports batched rollouts, enabling fast execution. Importantly, this framework is problem-agnostic, meaning it contains no problem-specific components, allowing it to be applied to a broader range of problems than those evaluated in this paper.

Our search framework consists of two main components: the improvement step function (illustrated in Figure 2) and the high-level augmented simulated annealing (ASA) algorithm (Algorithm 2). The improvement step function aims to enhance a given solution by iteratively applying the policy model through a series of deconstruction and reconstruction steps. The ASA algorithm integrates this function and supports batched execution for improved performance on the GPU. It is important to note that we parallelize solely on the GPU, requiring only a single CPU core during test time.

**Improvement Step**    The improvement step, the core component of the overall search algorithm, is depicted in Figure 2. The process begins with an initial solution $s_0$ that is passed to the policy DNN, which generates $K$ rollouts, each consisting of $M$ actions that specify the customers to be removed. Once the policy DNN completes its execution, these rollouts are sequentially applied to produce new candidate solutions. Specifically, the solution $s_0$ is first deconstructed based on the actions from the first rollout (yielding $\bar{s_0}$) and then reconstructed into $s_0'$. After reconstruction, a simulated annealing (SA) based acceptance criterion is used to determine whether $s_0'$ or $s_0$ should be retained, resulting in $s_1$. This process is repeated in each subsequent iteration. After $K$ iterations, the final solution $s_K$ is returned, representing the outcome of $K$ consecutive deconstruction and reconstruction operations. By performing these iterations sequentially, the solution $s_0$ is significantly modified, often leading to notable cost improvements between the initial input $s_0$ and the final output $s_K$.

**Augmented Simulated Annealing**    We introduced a novel simulated annealing (SA) algorithm to conduct a high-level search specifically designed for GPU-based parallelization. While parallel SA algorithms have been proposed in prior work, (Ferreiro et al., 2013; Jeong & Kim, 1990; Onbaşoğlu & Özdamar, 2001), their main concern is on the information exchange between CPU or GPU cores. In contrast, our approach focuses on executing parallel rollouts of the policy network on the GPU.

At a high level, the ASA technique, shown in Algorithm 2, modifies solutions over multiple iterations using a temperature-based acceptance criterion. This criterion allows worsening solutions to be accepted with a certain probability, which depends on the current temperature. The temperature $\lambda$ is manually set at the start of the search (line 2) and is gradually reduced after each iteration (line 15), resulting in a decreasing probability of accepting worsening solutions during the improvement step (line 6). For a detailed discussion on SA, we refer the reader to Gendreau et al. (2010).

To enable parallel search for a single instance, we employ the augmentation technique introduced in Kwon et al. (2020), which creates a set of augmentations $l_1', l_2', \ldots, l_A'$ for an instance $l$. The search is then conducted in parallel for these augmentations. After each modification by the improvement step procedure (line 6), solutions can be exchanged between different augmentations. Specifically,

---

**Algorithm 2** Augmented Simulated Annealing

---

1: **procedure** SEARCH(Instance $l$, Number of iterations $maxIter$, number of augmentations $A$, number of rollouts $K$, start temperature $\lambda^{start}$, temperature decay rate $\lambda^{decay}$, trained policy network $\pi_\theta$, threshold factor $\delta$)
2:     $\lambda \leftarrow \lambda^{start}$
3:     $\{l_1', l_2', \ldots, l_A'\} \leftarrow$ CREATEAUGMENTATIONS($l$)
4:     $s_a \leftarrow$ GENERATESTARTSOLUTION($l_a'$)    $\forall a \in \{1, \ldots, A\}$
5:     **for** $iter = 1, \ldots, maxIter$ **do**
6:         $s_a \leftarrow$ IMPROVEMENTSTEP($s_a, \pi_\theta, \lambda, K$)    $\forall a \in \{1, \ldots, A\}$
7:         $cost_a \leftarrow$ OBJ($s_a$)    $\forall a \in \{1, \ldots, A\}$
8:         $cost^* \leftarrow \min(cost_0, \ldots, cost_A)$
9:         $thresh \leftarrow cost^* + (\lambda \times \delta)$
10:         **for** $a = 1, \ldots, A$ **do**
11:             **if** $cost_a > thresh$ **then**
12:                 $s_a \leftarrow$ RANDOMCHOICE($\{s' \in \{s_0, \ldots s_A\} \mid$ OBJ($s'$) $< thresh\}$)
13:             **end if**
14:         **end for**
15:         $\lambda \leftarrow$ REDUCETEMPERATURE($\lambda, \lambda^{decay}$)
16:     **end for**
17: **end procedure**

---

we iterate over all augmentation instances (lines 10 to 14) and replace solutions that surpass a certain cost threshold with randomly selected solutions whose costs fall below the threshold. This threshold is calculated based on the cost of the current best solution and the temperature, adjusted by a factor $\delta > 1$, as shown in line 9. The goal is to replace solutions that are unlikely to surpass the quality of the current best solution, given the current temperature.

## 4 EXPERIMENTS

We evaluate NDS on three VRP variants with 100 to 2000 customers and compare to state-of-the-art learning-based and traditional OR methods. Additionally, we provide ablation experiments for the individual components of NDS and evaluate the generalization across different instance distributions. All experiments are conducted on a research cluster utilizing a single Nvidia A100 GPU per run. We will release our implementation of NDS, along with the instance generators, under an open-source license upon acceptance.

### 4.1 PROBLEMS

**CVRP**    The CVRP is one of the most extensively studied variants of the VRP. The goal is to determine the shortest routes for a fleet of vehicles tasked with delivering goods to a set of $N$ customers. Each vehicle begins and ends its route at a depot and is constrained by a maximum carrying capacity. We use the instance generator from Kool et al. (2019) to create scenarios with uniformly distributed customer locations, and the generator from Queiroga et al. (2022) for generating more realistic instances, with clustered customer locations to better simulate real-world conditions.

**VRPTW**    The VRPTW extends the traditional CVRP by adding time constraints for customer deliveries. Each customer has a time window, defining the earliest and latest allowable delivery times. Vehicles can arrive early but must wait until the window opens, adding scheduling complexity. All routes start at a central depot, with a fixed service duration for deliveries and travel times based on the Euclidean distance. The objective is to minimize the total travel time while respecting both vehicle capacity and time windows, making VRPTW more complex than the standard CVRP. To generate customer locations and demands, we use the CVRP instance generator from Queiroga et al. (2022), while time windows are generated following the methodology outlined by Solomon (1987).

**PCVRP**    The PCVRP is a variant of the VRP in which not all customers need to be visited. Each customer is associated with a prize, and the objective is to minimize the total travel cost minus the sum of collected prizes. Similar to the CVRP, all vehicles start and end their routes at a central depot and are constrained by vehicle capacities. To generate PCVRP instances, we use the instance

generator from Queiroga et al. (2022) to create customer locations and demands. Customer prize values are generated at random, with higher prizes assigned to customers with greater demand, reflecting the increased resources required to service them.

## 4.2 Search Performance

**Baselines**  We compare NDS to several heuristic solvers, including HGS (Vidal, 2022), SISRs (Christiaens & Vanden Berghe, 2020), and LKH3 (Helsgaun, 2017). Additionally, we include PyVRP (Wouda et al., 2024) (version 0.9.0), which is an open-source extension of HGS for other VRP variants. For the CVRP, we further compare NDS to the state-of-the-art learning-based methods, SGBS-EAS (Choo et al., 2022), BQ (Drakulic et al., 2023), LEHD (Luo et al., 2023), and GLOP (Ye et al., 2024b).

**NDS Training**  For each problem and problem size, we perform a separate training run. Training consists of 2000 epochs for settings with 1000 or fewer customers. For the 2000 customer setting, we resume training from the 1000 customer model checkpoint at 1500 epochs and train for an additional 500 epochs. In each epoch, we process 1500 instances, with each instance undergoing 100 iterations, 128 rollouts, and 10 initial improvement steps. The learning rate is set to $10^{-4}$ and 15 customers are selected per deconstruction step across all problem sizes. The training durations are approximately 5, 8, 15, and 8 days for the problem sizes 100, 500, 1000 and 2000, respectively. The training curves are presented in Appendix A, while visualizations of policy rollouts are available in Appendix B.

**Evaluation Setup**  At test time, we limit the runtime to 5, 60, 120, and 240 seconds of wall time per instance for HGS, SISRs, and NDS to ensure a fair comparison, as these methods process test instances sequentially. SGBS-EAS and LEHD, which process instances in batches, are given an equivalent search budget per batch. All approaches are restricted to using a single CPU core. For the CVRP, we use the test instances from Kool et al. (2019) for $N$=100 (10,000 instances), Drakulic et al. (2023) for $N$=500 (128 instances), and Ye et al. (2024b) for $N$=1000 and $N$=2000 (100 instances each). For the VRPTW and PCVRP, we generate new test sets consisting of 10,000 instances for $N$=100 and 250 instances for settings with more than 100 customers.

**NDS Test Configuration**  For NDS, the starting temperature $\lambda^{start}$ is set to 0.1 and decays exponentially to 0.001 throughout the search. The threshold factor $\delta$ is fixed at 15. During the improvement step, 200 rollouts are performed per instance, and each deconstructed solution is reconstructed 5 times ($1\times$ based on the selected order of the DNN and $4\times$ using a random customer order). The number of augmentations is set to 8 for the CVRP and VRPTW, and 128 for the PCVRP.

**Results**  Table 1 presents the performance of all compared methods on the test data. The gap is reported relative to HGS for the CVRP, and to PyVRP-HGS for the VRPTW and PCVRP. Across the 12 test settings, NDS delivers the best performance in 11 cases, with HGS being the only approach able to outperform it on the CVRP with 100 customers. Compared to other learning-based methods, NDS shows significant performance improvements across all CVRP sizes. On the CVRP with 2000 customers, NDS achieves a 7 percentage point improvement over the best-performing learning-based method, LEHD, and a 12 percentage point improvement over GLOP. Against the state-of-the-art HGS and its extension, PyVRP-HGS, NDS performs especially well on larger instances, achieving a gap of more than 2% across all problems for instances with 2000 customers. For the PCVRP, NDS also attains substantial gaps relative to PyVRP-HGS, exceeding 4% on instances with 500 or more nodes. When compared to SISRs, NDS maintains a small advantage on larger instances and demonstrates significantly better performance on small instances.

## 4.3 Ablation Studies

We perform a series of ablation experiments to assess the importance of different components of NDS. These experiments are conducted on separate validation instances with $N$=500 customers. The parameter configuration remains identical to the previous section, except the training is reduced to 1,000 epochs and the ASA search is limited by the number of iterations. For the CVRP and VRPTW, we run 1,000 iterations using 8 augmentations, while for the PCVRP, we perform 50 iterations with 128 augmentations.

Table 1: Performance on test data. The gap is calculated relative to HGS for the CVRP and relative to PyVRP-HGS for the VRPTW and PCVRP. Runtime is reported on a per-instance basis in seconds.

| | Method | N=100 | | | N=500 | | | N=1000 | | | N=2000 | | |
|---|---|---|---|---|---|---|---|---|---|---|---|---|---|
| | | Obj. | Gap | Time | Obj. | Gap | Time | Obj. | Gap | Time | Obj. | Gap | Time |
| **CVRP** | HGS | 15.57 | - | 5 | 36.66 | - | 60 | 41.51 | - | 121 | 57.38 | - | 241 |
| | SISRs | 15.62 | 0.32% | 5 | 36.65 | 0.01% | 60 | 41.14 | -0.83% | 120 | 56.04 | -2.27% | 240 |
| | LKH3 | 15.64 | 0.50% | 41 | 37.25 | 1.66% | 174 | 42.16 | 1.61% | 408 | 58.12 | 1.35% | 1448 |
| | SGBS-EAS | 15.59 | 0.17% | 5 | - | - | - | - | - | - | - | - | - |
| | BQ (BS64) | 15.74 | 1.13% | 1 | 37.51 | 2.32% | 23 | 43.32 | 4.36% | 164 | - | - | - |
| | LEHD (RRC) | 15.61 | 0.30% | 5 | 37.04 | 1.04% | 60 | 42.47 | 2.31% | 121 | 60.11 | 4.76% | 246 |
| | GLOP (LKH3) | - | - | - | - | - | - | 45.90 | 10.58% | 4 | 63.00 | 9.79% | 6 |
| | NDS | 15.57 | 0.04% | 5 | 36.57 | **-0.20%** | 60 | 41.11 | **-0.90%** | 120 | 56.00 | **-2.34%** | 240 |
| **VRPTW** | PyVRP-HGS | 12.98 | - | 5 | 49.01 | - | 60 | 90.35 | - | 120 | 173.46 | - | 240 |
| | SISRs | 13.00 | 0.20% | 5 | 48.09 | -1.87% | 60 | 87.68 | -2.98% | 120 | 167.49 | -3.49% | 240 |
| | NDS | 12.95 | **-0.19%** | 5 | 47.94 | **-2.17%** | 60 | 87.54 | **-3.14%** | 120 | 167.48 | **-3.50%** | 240 |
| **PCVRP** | PyVRP-HGS | 10.11 | - | 5 | 44.97 | - | 60 | 84.91 | - | 120 | 165.56 | - | 240 |
| | SISRs | 9.94 | -1.66% | 5 | 43.22 | -3.90% | 60 | 81.12 | -4.55% | 120 | 158.17 | -4.54% | 240 |
| | NDS | 9.90 | **-2.07%** | 5 | 43.12 | **-4.12%** | 60 | 80.99 | **-4.71%** | 121 | 158.09 | **-4.60%** | 241 |

Table 2: Ablation experiments.

(b) Insertion order

| Order | CVRP | VRPTW | PCVRP |
|---|---|---|---|
| DNN+Random | **36.81** | **47.68** | **42.96** |
| Random | 36.86 | 47.76 | 43.05 |

(a) Impact of the message passing layer (MPL) and the tour encoding layer (TEL) on performance.

| MPL | TEL | CVRP | VRPTW | PCVRP |
|---|---|---|---|---|
| ✓ | ✓ | 36.81 | **47.68** | **42.96** |
| ✓ | ✗ | 36.82 | 47.75 | 43.13 |
| ✗ | ✓ | 36.81 | 47.74 | 42.98 |
| ✗ | ✗ | 36.87 | 47.87 | 43.62 |

(c) Deconstruction policy

| Policy | CVRP | VRPTW | PCVRP |
|---|---|---|---|
| DNN | **36.81** | **47.68** | **42.96** |
| Heuristic | 37.03 | 48.16 | 43.61 |

**Network Architecture**    We assess the impact of the message passing layer (MPL) and tour encoding layer (TEL) on overall performance by training separate models without these components. Table 2a summarizes the resulting search performance. Excluding both layers leads to a significant performance drop, with a 1.5% reduction on the PCVRP. Even the removal of a single layer causes a notable performance decline, particularly for the VRPTW and PCVRP. The VRPTW in particular benefits from both layers, likely due to the MPL's ability to better interpret and handle time windows.

**Insertion Order**    The insertion algorithm reinserts removed customers in a specified order. During testing, we reconstruct a deconstructed solution five times using different customer orders and retain the best solution. For the first reconstruction iteration, we use the customer order provided by the DNN, while for the remaining four iterations we use a random order. We compare our standard setting to using only random orderings for all five insertion iterations to assess whether the ordering enhances overall search performance. The results in Table 2b show that using a only random orderings leads to significantly worse performance across all three problems, indicating that the learned policy not only plays a crucial role in deconstruction, but also significantly influences reconstruction.

**Learned Policy**    We assess the relevance and effectiveness of the learned deconstruction policy by replacing it with a handcrafted heuristic based on the destroy procedure outlined in Christiaens & Vanden Berghe (2020). The resulting approach eliminates any learned components, but is otherwise identical to NDS. The performance comparison, shown in Figure 2c, reveals that the heuristic deconstruction policy performs significantly worse than the learned counterpart, with performance gaps of up to 1.5% on the PCVRP. This demonstrates that the DNN is capable of learning a highly efficient policy that surpasses handcrafted methods in this use case.

Table 3: Out-of-distribution (OOD) vs. in-distribution (ID) performance on the CVRP500.

| Method | Uniform Locations | | | | | | Clustered Locations | | | | | |
| | Low Capacity | | | High Capacity | | | Low Capacity | | | High Capacity | | |
| | Obj. | Gap | Time | Obj. | Gap | Time | Obj. | Gap | Time | Obj. | Gap | Time |
|---|---|---|---|---|---|---|---|---|---|---|---|---|
| HGS | 91.73 | - | 60 | 47.89 | - | 60 | 88.20 | - | 60 | 44.53 | - | 61 |
| SISRs | 91.34 | -0.38% | 60 | 47.79 | -0.17% | 60 | 87.78 | -0.48% | 60 | 44.31 | -0.49% | 60 |
| NDS (OOD) | 91.15 | -0.59% | 60 | 47.70 | -0.36% | 60 | 87.75 | -0.53% | 60 | 44.29 | -0.54% | 60 |
| NDS (ID) | 91.14 | -0.59% | 60 | 47.69 | -0.38% | 60 | 87.70 | -0.58% | 60 | 44.26 | -0.60% | 60 |

## 4.4 GENERALIZATION

One major advantage of learning-based solution approaches is their ability to adapt precisely to the specific type of instances at hand. However, in real-world scenarios, concept drift in the instance distributions cannot always be avoided. In this experiment, we evaluate whether the learned policies of NDS can handle instances sampled from a slightly different distribution. For the CVRP with $N$=500, we train a policy on instances with medium-capacity vehicles and customer locations that follow a mix of uniform and clustered distributions. We then evaluated the learned policy on instances with low- and high-capacity vehicles, and customer locations following either uniform or clustered distributions. Additionally, we train distribution-specific models for each test setting for comparison. As a baseline, we compare against HGS and SISRs, giving all approaches the same runtime. The results are shown in Table 3, where NDS (OOD) represents the model's performance when the training and test distributions differ, and NDS (ID) represents the setting where the training and test distributions are identical. Overall, the performance difference between the two settings is minimal, indicating that NDS generalizes well across different distributions. Interestingly, the distribution of customer locations has a larger impact on performance than vehicle capacity.

## 4.5 SCALABILITY ANALYSIS

We assess the scalability of NDS by analyzing its runtime and GPU memory consumption on CVRP instances of varying sizes. Figure 3 presents the relative resource usage as a function of problem size. Overall, NDS demonstrates strong scalability to larger instances. Notably, solving instances with 1,000 customers requires only 61% more runtime and 23% more memory compared to instances with 100 customers, despite the problem size increasing by an order of magnitude.

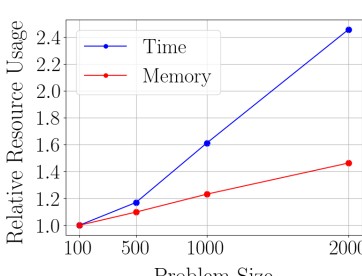

Figure 3: Scalability

## 5 CONCLUSION

In this work, we introduced a novel search method, NDS, which leverages a learned policy to deconstruct solutions for routing problems. NDS presents several key advantages. First, it delivers superior performance, consistently outperforming state-of-the-art OR methods under equal runtime. Second, NDS scales effectively to larger problem instances, handling up to $N$=2000 customers, due to the fact that the number of customers selected by the policy is independent of the problem size. Third, it demonstrates strong generalization across different data distributions. Finally, NDS is easily adaptable to new vehicle routing problems, requiring only small adjustments to the greedy insertion heuristic and the model input.

A notable limitation is the reliance on a GPU for executing the policy network. Future research could explore model distillation techniques to lower the computational requirements or investigate whether the underlying principles of the learned policies can be approximated using faster, more efficient algorithms.

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

## A  TRAINING CURVES

Figure 4 presents the training curves for all experiments conducted across the three problem types and four problem sizes. Note that the training of the models for $N$=2000 is warm-started using the model weights from $N$=1000 after 1,500 epochs.

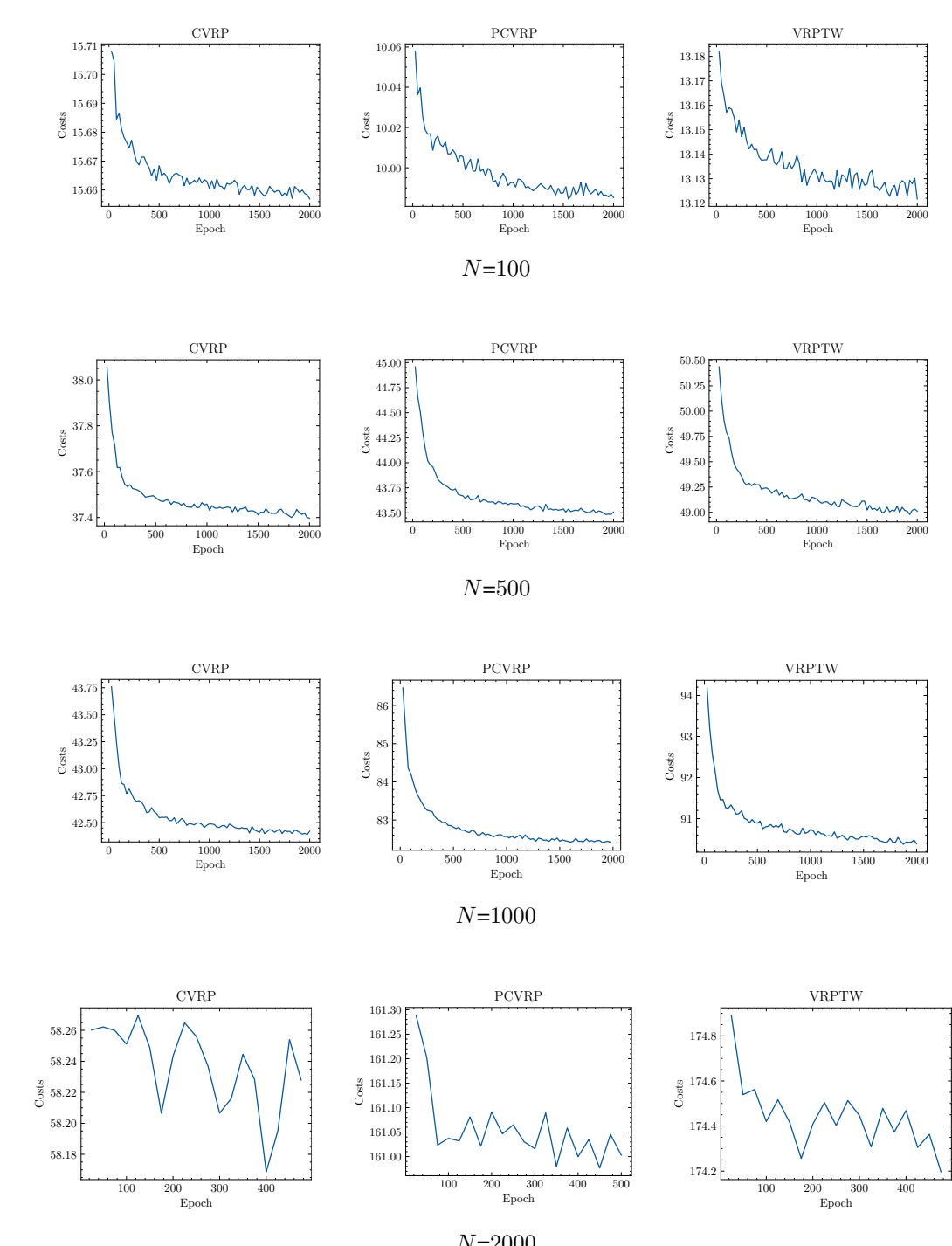

Figure 4: Training curves.

# B VISUALIZATIONS OF POLICY ROLLOUTS

Figures 5, 6, and 7 show visualizations of different policy rollouts for the CVRP, PCVRP, and VRPTW, respectively. For each problem, we display two different instances, and for each instance, six rollouts are shown. Customers selected for deconstruction are circled in red. We note that the nodes selected for each deconstruction differs, sometimes significantly, allowing NDS to try out a variety of options in each iteration.

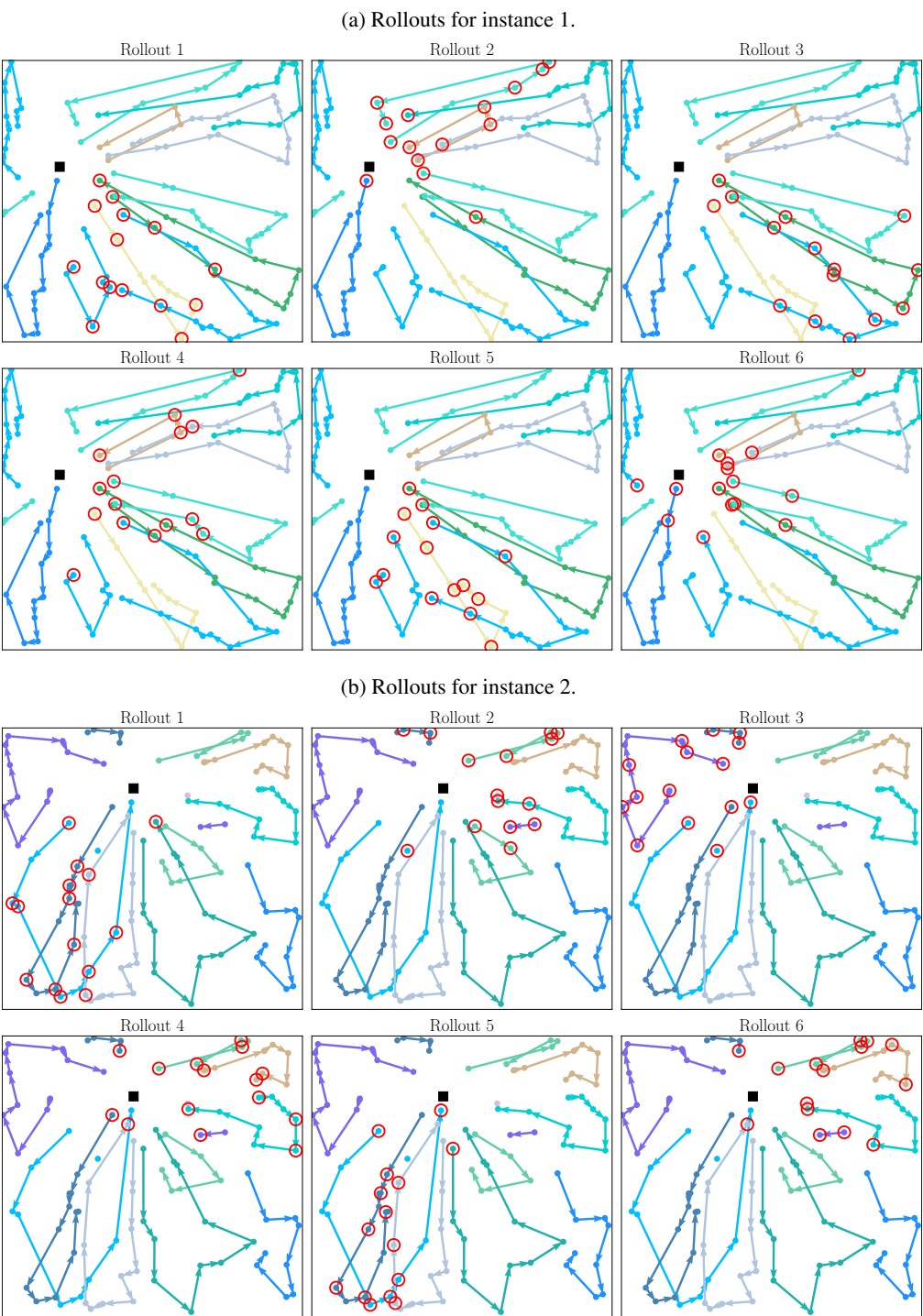

Figure 5: Rollouts for two selected instances for the CVRP with $N=100$ (best viewed in color).

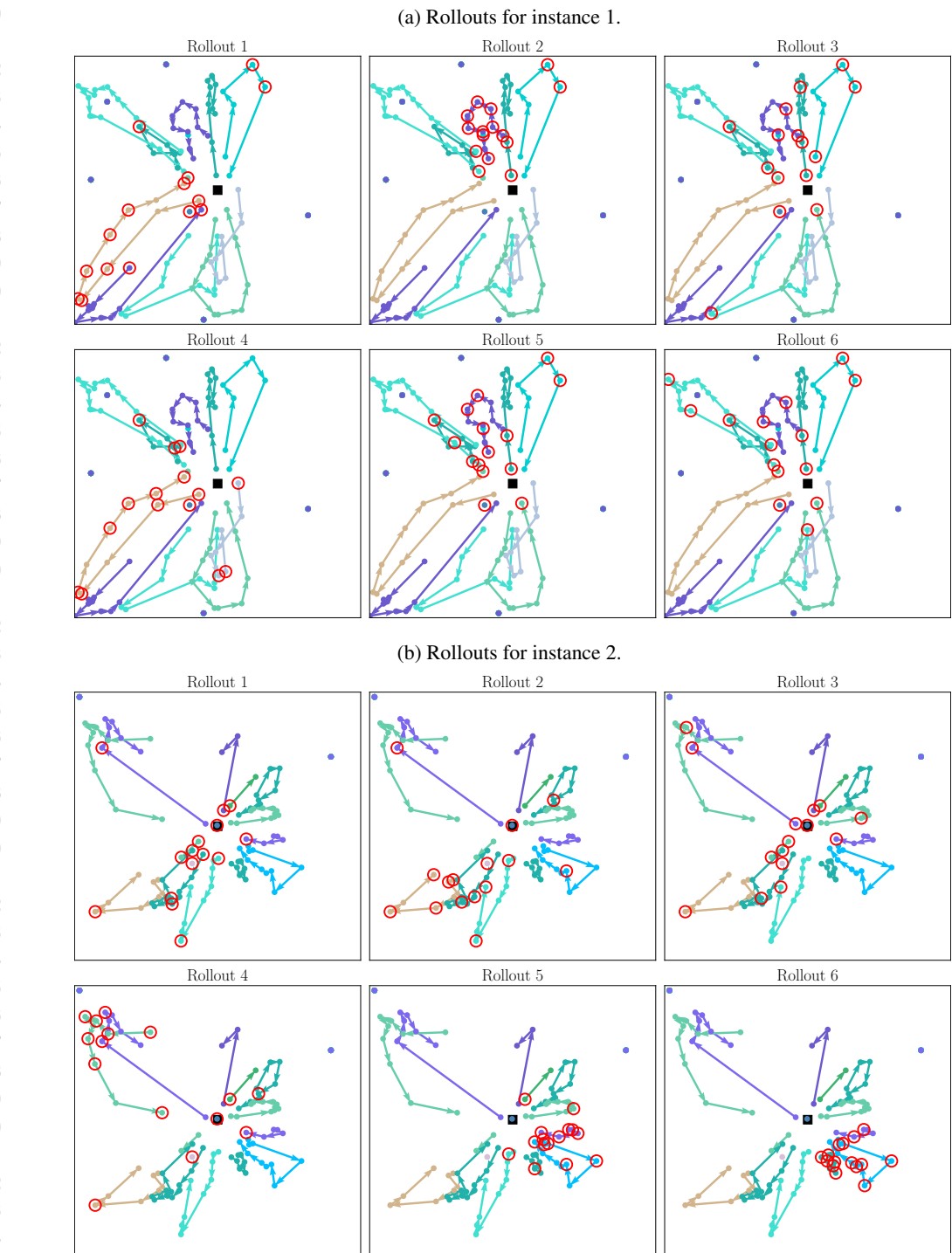

Figure 6: Rollouts for two selected instances for the PCVRP with $N$=100 (best viewed in color).

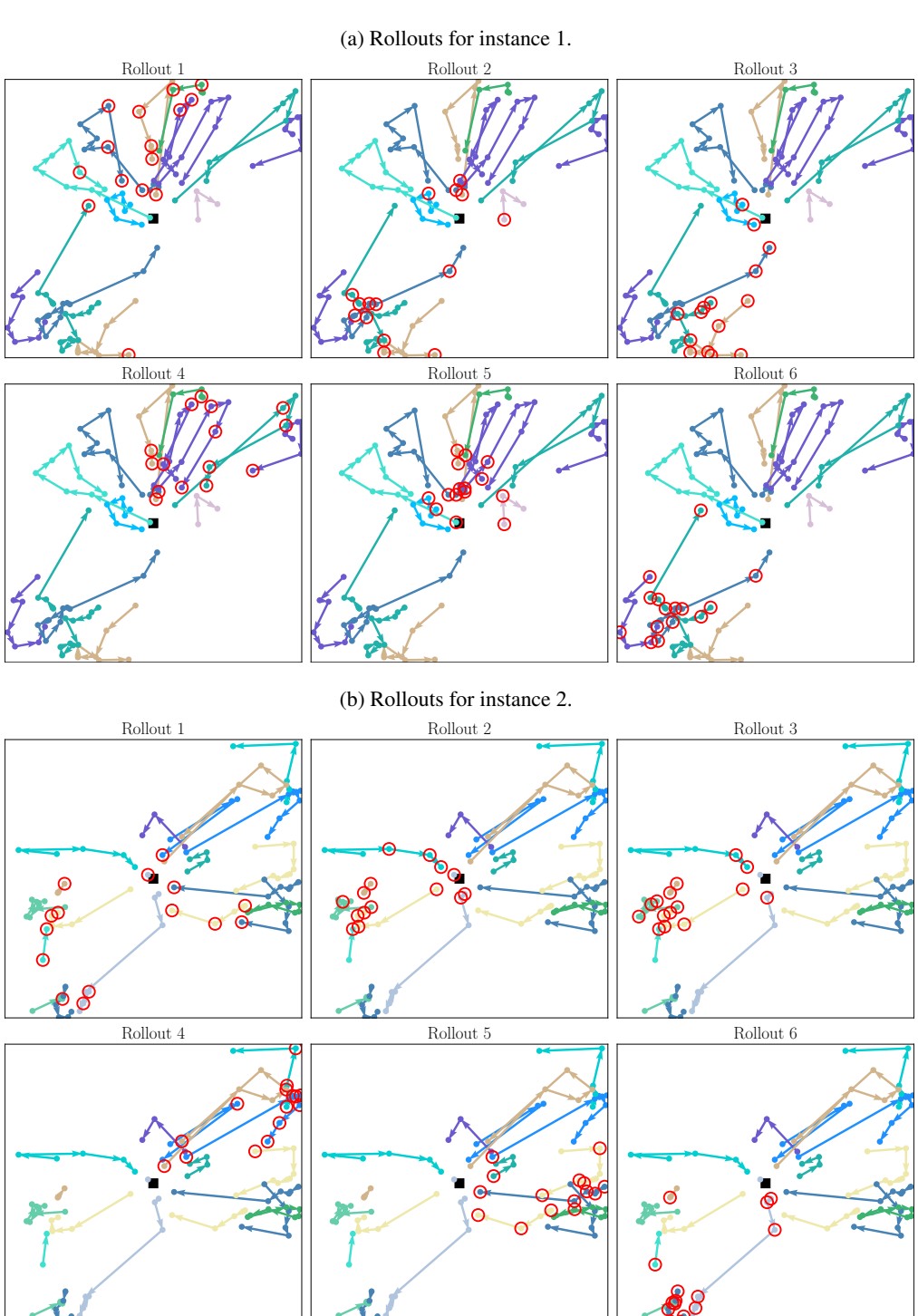

Figure 7: Rollouts for two selected instances for the VRPTW with $N$=100 (best viewed in color).

