# OpenReview forum: "Neural Deconstruction Search for Vehicle Routing Problems"
_ICLR.cc/2025/Conference — Submitted to ICLR 2025_

### Official Review · Reviewer_CuKB · 2024-10-28

**Soundness:** 3
**Presentation:** 2
**Contribution:** 2
**Rating:** 3
**Confidence:** 5

**Summary:**

Tins paper uses deep neural network (DNN) to trun the operation of removing nodes into a sequence generation process with the aim of achieving a fast serach process without sacrificing the high quality search guidance of the DNN. The solution is then finally reconstructed using greedy methods. While the work has some merit, the main contribution does not meet the novelty expectations of ICLR.

**Strengths:**

1. Although the method is simple, it achieves a certain effect.

**Weaknesses:**

1.	Just replacing a heuristic process of removing nodes with a DNN is too simple. The contribution here does not significantly advance the field. I think the authors need to explain what are the differences and advantages of the proposed method for removing nodes over the previous methods?
2.	A similar approach of using DNNs to remove and reconstruct solutions has been used in papers [1]. However the proposed method only uses DNNs for node removal and greedy methods for reconstruction. There are three other papers[2-4] on learning similar local search operations in VRP, and those methods can also be used directly to learn removal and reconstruction.
3.	IMPROVEMENTSTEP is used first in Algorithm 1, with no specific explanation as to why.
4.	There are no details about the way you use DNNs in the decoding process, leading the reader to have little understanding of the mechanism.
5.	This part of the formulation needs further discussion: “This contrasts with construction-based methods, where each decision is independent of prior selections.”. However, the current construction-based methods also retain the customer information selected in the previous step or part of the path information during the decoding process.

[1]Efficient Neural Neighborhood Search for Pickup and Delivery Problems. IJCAI 2022.

[2]Learning to Iteratively Solve Routing Problems with Dual-Aspect Collaborative Transformer. NeurIPS 2021.

[3] Learning Improvement Heuristics for Solving Routing Problems. TNNLS 2021.

[4] Learning to Search Feasible and Infeasible Regions of Routing Problems with Flexible Neural k-Opt. NeurIPS 2023.

**Questions:**

What are the advantages of this paper over previous papers that have implemented reconstructive solving using DNN, please explain in detail.

---

> ### Author Response · Authors · 2024-11-22
>
> Thank you for reviewing our paper and providing feedback!
>
> > Just replacing a heuristic process of removing nodes with a DNN is too simple. The contribution here does not significantly advance the field. I think the authors need to explain what are the differences and advantages of the proposed method for removing nodes over the previous methods?
>
> We do not just replace a single component of an already existing method. NDS as a whole is a completely new large neighborhood search method. All components of NDS have been designed with a GPU-accelerated search in mind.
>
> We highlight the advantages of NDS in the conclusion as follows: "NDS presents several key advantages. First, it delivers superior performance, consistently outperforming state-of-the-art OR methods under equal runtime. Second, NDS scales effectively to larger problem instances, handling up to N=2000 customers, due to the fact that the number of customers selected by the policy is independent of the problem size. Third, it demonstrates strong generalization across different data distributions. Finally, NDS is easily adaptable to new vehicle routing problems, requiring only small adjustments to the greedy
> insertion heuristic and the model input."
>
> We also take issue with the reviewer implying that a simple mechanism is not sufficient for publication at a top conference. Quite the contrary, simple is good. Does the reviewer only want complicated methods?
>
> Consider previous works at top conferences, such as the POMO method at NeurIPS (Kwon et al (2020)). This method introduces two simple ideas, instance augmentation and a modified loss function. That is it -- and it is NeurIPS paper. It was accepted because (1) simple is good and (2) it improves performance. NDS has a relatively simple idea (admittedly the details are not so simple), and it works. If any of the details are unclear, we would be happy to try to clear them up and thank the reviewer for their constructive input.
>
>
> > A similar approach of using DNNs to remove and reconstruct solutions has been used in papers [1]. However the proposed method only uses DNNs for node removal and greedy methods for reconstruction. There are three other papers[2-4] on learning similar local search operations in VRP, and those methods can also be used directly to learn removal and reconstruction.
>
> There are significant differences between NDS and these methods. Most of the methods make only small changes to a solution at each step (considering only a very small neighborhood), in contrast NDS is a large neighborhood search method that makes significant changes at each step. Furthermore, these method reinsert with a neural network [1] or swap edges directly [2-4]. In contrast, we focus on only learning the deconstruction and use a greedy insertion algorithm for reconstruction.
>
> > IMPROVEMENTSTEP is used first in Algorithm 1, with no specific explanation as to why.
>
> We write in line 186: "By refining s with NDS’s main search component before the training rollouts, we ensure that the training focuses on improving non-trivial solutions."
>
>
> > There are no details about the way you use DNNs in the decoding process, leading the reader to have little understanding of the mechanism.
>
> Thank you for pointing this out. We will try to make this more clear in Section 3.3.
>
> > This part of the formulation needs further discussion: “This contrasts with construction-based methods, where each decision is independent of prior selections.”. However, the current construction-based methods also retain the customer information selected in the previous step or part of the path information during the decoding process.
>
> For construction approaches the order in which previous customers have been visited is irrelevant when deciding where to go next. We will rephrase the sentence to make this more clear.

---

> > ### Comment · Reviewer_CuKB · 2024-11-24
> >
> > Thanks to the authors' efforts in the rebuttal. For now, however, I lean to maintain my score. The reasons are as follows:
> > 1. The authors emphasize that " Furthermore, these method reinsert with a neural network or swap edges directly. In contrast, we focus on only learning the deconstruction and use a greedy insertion algorithm for reconstruction." and "All components of NDS have been designed with a GPU-accelerated search in mind." which are limited contribution in my opinion.
> > 2. Too much emphasis on experimental results rather than methodological analysis.

---

### Official Review · Reviewer_1gVo · 2024-10-31

**Soundness:** 2
**Presentation:** 2
**Contribution:** 1
**Rating:** 3
**Confidence:** 4

**Summary:**

This paper adopts a destroy-reconstruct iterative search framework to improve the quality of VRP solutions. The destruction step is performed by the proposed attention-based model, which removes customer nodes step by step from the solution; then, the removal of customer nodes are inserted into the destroyed solution node by node using a simple greedy insertion method. In addition, this paper also adopts a simulated annealing mechanism to allow worsening solutions to be accepted during the iteration process. Experimental results show that the proposed method achieves state-of-the-art performance on CVRP, CVRPTW, and PCVRP.

**Strengths:**

1. The paper writing is fluent.
2. Experiments show outstanding performances on three VRPs.

**Weaknesses:**

1. From my point of view, the framework of this paper is mainly based on SISRs. This paper replaces the heuristic destruction process of SISRs with a learning-based destruction strategy. The greedy insertion with simulated annealing in this paper is essentially the same as the heuristic reconstruction process "greedy insertion with blinks" in SISRs, both of which have the probability of accepting differential solutions to jump out of the local optimum. Based on the above observations, I think the framework of this paper remains at the engineering level. It is a migration of the framework of SISRs, which is not novel enough.
2. Since the performance of SISRs is promising, it is foreseeable that it will be better after adding the machine learning strategy. In Table 1, the performance gap between the proposed method and SISRs is not significant especially when N is large. Therefore, I believe that the main performance source of the proposed method is the framework of SISRs.
3. The proposed method is similar to the SISRs framework, so I suggest the authors discuss its relation to SISRs in the related work section. Many details are not explained clearly, such as how to train the model on VRPTW and PCVRP, what changes are made, and how to meet the constraints.
4. There needs more baselines to compare. It should introduce more comparison algorithms for VRPTW and PCVRP, and CVRP should also introduce newly emerged baselines such as ELG [1] and UDC [2].

[1] Gao,  Chengrui, et al. "Towards generalizable neural solvers for vehicle routing problems via ensemble with transferrable local policy." arXiv preprint arXiv:2308.14104 (2023).

[2] Zheng,  Zhi, et al. "Udc: A unified neural divide-and-conquer framework for large-scale combinatorial optimization problems." arXiv preprint arXiv:2407.00312 (2024).

**Questions:**

1. Why is greedy insertion used to reconstruct the solution, rather than other methods such as regret insertion?
2. Can the proposed method solve the TSP?
3. There is no ablation study on the random seed v. And could you describe the reason for choosing the hyper-parameters in this paper, such as threshold factor and temperature?
Minor question:
4. Figure 3 should be corrected to a vector diagram.

---

> ### Author Response · Authors · 2024-11-22
>
> Thank you for reviewing our paper and providing feedback!
>
> > From my point of view, the framework of this paper is mainly based on SISRs. This paper replaces the heuristic destruction process of SISRs with a learning-based destruction strategy. The greedy insertion with simulated annealing in this paper is essentially the same as the heuristic reconstruction process "greedy insertion with blinks" in SISRs, both of which have the probability of accepting differential solutions to jump out of the local optimum. Based on the above observations, I think the framework of this paper remains at the engineering level. It is a migration of the framework of SISRs, which is not novel enough.
>
> NDS and SISRs are both based on the large neighborhood search framework so there are naturally some similarities. However, both methods are still very different. As you write correctly, SISRs uses "greedy insertion with blinks". However, we do not use this mechanism during reconstruction and there is also no similarity of this mechanism to anything we do. Furthermore, the repair of SISRs is significantly more complex, as it uses different handcrafted ordering heuristics that order customers before insertion. We do not use this. We use a simple greedy insertion heuristic because it is literally the simplest heuristic to implement (except maybe random insertion).
>
> Even if our work were merely "a migration of the framework of SISRs," it would still be remarkable that we outperform the complex original SISRs destroy mechanism using a learned policy.
>
> We emphasize that the reviewer guidelines state "Submissions bring value to the ICLR community when they convincingly demonstrate new, relevant, impactful knowledge.".
>
> Before the NDS paper, it was not known that using DNNs in the way we use them could lead to such good performance. If it had been known, certainly someone would have published it. We encourage you to rethink your score with these guidelines in mind.
>
> > Since the performance of SISRs is promising, it is foreseeable that it will be better after adding the machine learning strategy.
>
> It is absolutely not foreseeable that adding a powerful ML components improves performance of a method. The computational expense of ML components has inhibited their widespread use in optimization techniques. If it was so easy, we would have nothing left to research by now. Consider that the ML community has mostly ignored SISRs until now.
>
>
> > There needs more baselines to compare. It should introduce more comparison algorithms for VRPTW and PCVRP, and CVRP should also introduce newly emerged baselines such as ELG [1] and UDC [2].
>
> Why do we need more baselines? Do you have any OR methods in mind that in your opinion will outperform our current baselines? We aim to include comparisons to ELG and UDC in the future, as these methods do show good performance, but we note that it ought to be clear from the results tables in both of those papers that ELG and UDC will not outperform NDS on the instances presented.
>
>
> > Why is greedy insertion used to reconstruct the solution, rather than other methods such as regret insertion?
>
> We chose greedy insertion because it is the most trivial reconstruction method we could think of (besides random insertion). We want to keep the non-learning based part of this method as simple and trivial as possible to allow for an easy adaption of the method to new problems. Regret insertion is of course a very interesting idea to try, and we thank the reviewer for the suggestion.
>
> > Can the proposed method solve the TSP?
>
> Yes, it could be adapted to the TSP. However, we recommend to use the Concorde or LKH for solving the TSP. Large neighborhood techniques have not traditionally beaten LKH on the TSP.
>
> > There is no ablation study on the random seed v. And could you describe the reason for choosing the hyper-parameters in this paper, such as threshold factor and temperature?
>
> We are considering including an additional ablation study. The parameters were chosen based on preliminary experiments, during which we tested a few carefully hand-selected values.
>
> > Figure 3 should be corrected to a vector diagram.
>
> Thank you. We will correct the figure.

---

> > ### Comment · Reviewer_1gVo · 2024-11-22
> >
> > Thanks for your rebuttal.
> >
> > >``Why do we need more baselines?``
> >
> > Generally speaking, different papers will have different experimental setups, and for me, responsible authors should show the superiority of the proposed methodology in the same setup to readers of a wider background.
> >
> > >``If it was so easy, we would have nothing left to research by now.``
> >
> > SISRs uses a heuristic destruction strategy so-called Adjacent string removal. This strategy is consistent for all scales and VRPs. The ML approach at least takes into account the scales and the specific VRP as a prior, naturally foreseeable that it will be better after adding the machine learning strategy.
> >
> > >``Before the NDS paper, it was not known that using DNNs in the way we use them could lead to such good performance.``
> >
> > I super agree that the existing community seems to have overlooked the efficiency of SISRs, and thank you for reminding the community of this with your work. NDS is indeed superior but has only a small increase in performance relative to SISRs. I think for any heuristic, it has been a community consensus for even years ago to consider learning methods to get some lift [1,2]. Well, it's also worth noting that these two methods train only one model for all scales.
> >
> > ***
> >
> > As a new question, I super recommend you add an experiment, running NDS with SISRs and HGS, for a long enough period to demonstrate the properties of NDS in terms of convergence curve and efficiency.
> >
> > Looking forward to your reply!
> >
> > [1] Xin, Liang, et al. “Neurolkh: Combining deep learning model with lin-kernighan-helsgaun heuristic for solving the traveling salesman problem.” Advances in Neural Information Processing Systems 34 (2021): 7472-7483.
> >
> > [2] Zheng, Jiongzhi, et al. “Combining reinforcement learning with Lin-Kernighan-Helsgaun algorithm for the traveling salesman problem.” Proceedings of the AAAI conference on artificial intelligence. vol. 35. no. 14. 2021.

---

### Official Review · Reviewer_BUBj · 2024-11-03

**Soundness:** 2
**Presentation:** 3
**Contribution:** 1
**Rating:** 3
**Confidence:** 5

**Summary:**

This paper falls within the realm of using learning-based approaches or elements to solve vehicle routing problems. Specifically, it proposes to deviate from the existing step-by-step approaches that usually focus on autoregressive construction techniques. Instead, it proposes an iterative search framework that uses a neural policy for deconstruction and a relatively simple greedy insertion algorithm to repair the respective solutions.
The paper shows promising results on various vehicle routing problem variants.

Overall, I think that this paper makes a valuable contribution to the field but is currently borderline in its exposition as the authors clearly oversell the contribution. Yet, this aspect can easily be healed and if the authors are willing to do so, I believe that this paper will be of interest to the ICLR community and could be accepted.

**Strengths:**

1) The paper is technically well-written and easy to follow

2) The presented experiments are of sufficient breadth

3) The proposed methodology opens an interesting avenue for algorithm design that deviates from existing approaches that usually focus on using neural policies to construct solutions in a step-by-step fashion.

4) The results show promising performance compared to existing methods.

**Weaknesses:**

As indicated in my summary, the authors are currently (significantly) overselling the contribution of the proposed method.

This relates to the fact that numerical experiments are - to some extent - comparing apples with oranges due to limiting the computation times of the studied algorithms instead of using a proper performance-based stopping criterion. This experimental design choice clearly favors the search technique proposed by the authors, as the neural deconstruction works instantaneously.

In practice, one would not use such a time-based criterion as the problems studied are static problems usually solved in a day-ahead fashion, where solution times are not limited to seconds. In such cases, one would usually finetune an algorithm based on a performance-based stopping criterion, i.e., the number of consecutive iterations without improvements.

Beyond this rather unconventional experimental design decision, the authors also do not provide details if and if so how the benchmark algorithms have been tuned.

Looking at the fact that hyperparameter tuning seems to be missing for the benchmark algorithms and that the experimental design in general favors the proposed algorithm, I think that statements like "Our approach surpasses the performance of state-of-the-art operations research methods" are not sufficiently substantiated and should be toned down, not only in the abstract but in the manuscript.

I think that the authors make an interesting contribution to the field that is obvious without such overselling statements. The paper as well as the understanding of the reader will benefit from a more nuanced discussion and analyses of performance and computational complexity. I think that the authors can easily address this aspect, e.g., by
1) toning down the respective claims, particularly in the papers abstract, contribution section, and results discussion.
2) adding information on hyperparameter tuning for all algorithms in an appendix
3) adding results where the benchmark algorithms have a more suitable stopping criterion. -even if the proposed algorithm then does not surpass the benchmarks, one can see the full picture and discuss the results for both stopping criteria more nuanced
4) commenting on the training effort of the proposed method compared to the benchmarks not requiring such a training phase

**Questions:**

In Section 4.4 you discuss generalization against distribution shifts, did you also investigate this for varying instance sizes?

---

> ### Author Response · Authors · 2024-11-22
>
> Thank you for reviewing our paper and providing extensive feedback!
>
> > This relates to the fact that numerical experiments are - to some extent - comparing apples with oranges due to limiting the computation times of the studied algorithms instead of using a proper performance-based stopping criterion. This experimental design choice clearly favors the search technique proposed by the authors, as the neural deconstruction works instantaneously.
> In practice, one would not use such a time-based criterion as the problems studied are static problems usually solved in a day-ahead fashion, where solution times are not limited to seconds. In such cases, one would usually finetune an algorithm based on a performance-based stopping criterion, i.e., the number of consecutive iterations without improvements.
>
> We acknowledge disagreements between communities about whether one should investigate iterations or runtime. However, it is important to note that runtime is the accepted criterion for routing problems in the entire operations research community and much of the work within AI. The paper associated with the well-known HGS solver [1] used a similar "unconventional experimental design" setup where all methods are limited by runtime. Can you point us to any central paper of the field that argues that runtimes are always irrelevant when solving VRP variants?
>
> Let us also note that we are addressing the vehicle routing problem as an operational problem, not as a strategic or tactical problem. We view this in the context of human-in-the-loop, interactive decision support. This means solutions cannot be solved the day before. They must be solved fast, and the solutions are then displayed to users who may incorporate further constraints into the model and re-solve. While this work is not at the stage where we can implement it in an interactive system, we are proud to have achieved the necessary speed.
>
>
> > Beyond this rather unconventional experimental design decision, the authors also do not provide details if and if so how the benchmark algorithms have been tuned.
> Looking at the fact that hyperparameter tuning seems to be missing for the benchmark algorithms and that the experimental design in general favors the proposed algorithm, I think that statements like "Our approach surpasses the performance of state-of-the-art operations research methods" are not sufficiently substantiated and should be toned down, not only in the abstract but in the manuscript.
>
> We have not performed hyperparameter optimization on either NDS or any of the baseline approaches. We note that the baselines have likely undergone tuning by their respective authors. Tuning all methods for the different scenarios is very computationally expensive. Both our main baselines HGS and SISRs have been designed and evaluated on instances very similar to ours as we use the instance generator from [2] and would likely not benefit much from tuning. We note that it is not standard to tune all baseline methods in the ML community due to the high computational costs associated with tuning a method that requires GPUs.
>
>
> [1] Vidal, Thibaut. "Hybrid genetic search for the CVRP: Open-source implementation and SWAP* neighborhood." Computers & Operations Research 140 (2022): 105643. \
> [2] Eduardo Queiroga, Ruslan Sadykov, Eduardo Uchoa, and Thibaut Vidal. 10,000 optimal CVRP solutions for testing machine learning based heuristics. In AAAI-22 Workshop on Machine Learning for Operations Research (ML4OR), 2022.
>
> > As indicated in my summary, the authors are currently (significantly) overselling the contribution of the proposed method.
>
> Based on our responses above, we honestly do not believe that we oversell our method. Does NDS need a costly GPU at test time? Yes. Does it need an expensive training phase? Yes. Does it dominate other approaches across all possible runtime limits? Likely not (but which method does?) Does it surpasses the performance of state-of-the-art operations research methods across three challenging vehicle routing problems of various problem sizes in our experiments? Yes! And this is a rather exciting result, one that we have been working towards for many years.
>
>
> > In Section 4.4 you discuss generalization against distribution shifts, did you also investigate this for varying instance sizes?
>
> No, currently we focus on distribution shifts only. We are currently considering showing the generalization ability of a model trained on instances with 1000 customers when applied to instances with 2000 customers.

---

> > ### Comment · Reviewer_BUBj · 2024-11-22
> >
> > I thank the authors for commenting on all of my comments.
> >
> > While I appreciate that the authors are excited about their work, I am surprised at how they react to constructive criticism.
> >
> > Let me just reiterate a few points:
> > 1) I strongly disagree with the authors' statement that "runtime is the accepted criterion for routing problems in the entire operations research community". The contrary is the case: the whole operations research community agrees that for most algorithms, a trade-off often exists between runtime and solution quality. If the authors screen papers published on VRPs, they will find two types: i) papers that improve on solution quality with runtime being a secondary objective. Many papers exist in top-tier journals like Transportation Science with worse runtimes but better solution quality; ii) papers that improve compared to existing papers upon runtime without or by only slightly worsening solution quality. It is unclear how one can support the authors' statement based on such observations.
> >
> > 2) Even for human-in-the-loop problems, one might be willing to wait longer than a few seconds to interact with the algorithm. If not, the authors must present their work in such a context and show that swift interactions are beneficial in this case. I think the paper is far from even touching upon such a setting as acknowledged by the authors.
> >
> > 3) I agree with the authors that "probably the algorithms have been tuned," but in a different setting under different boundary conditions and with different scope. Avoiding retuning in this case adds up more to the initially mentioned comparison of apples and oranges.
> >
> > 4) I am wondering why the authors are so defensive on their assumptions. One could have easily rerun some experiments and reported solution quality over different runtimes by simply saving the solutions after different time limits during one run, which would have allowed either for an even more substantial contribution statement if the proposed method finds new BKS or outperforms the existing methods after longer runtimes, or, alternatively, would have allowed for a more nuanced results discussion and contribution statement.
> > Looking at the fact that the authors avoid such a comparison does not increase my trust in the claims made.
> >
> > 5) I will not comment on the oversold contribution again - this is probably a hopeless debate, looking at the fact that the authors disagree with all of my comments.
> >
> > I gave my initial score, looking at the paper's potential and anticipating that the authors would address some of my comments. As this does not seem to be the case, I do not think this paper should be published at ICLR in its current form, and I will adjust my score accordingly.
> >
> > If the authors are that confident about their experimental design decisions and their acceptance in the operations research community, they should probably submit their work to an outlet in this field. Still, I am rather confident that as long as they submit to a top journal, they will receive similar comments and concerns in this community as well.
> >
> > This being said, I wish the authors the best of luck in publishing their work in a suitable venue.

---

> > > ### Author Response · Authors · 2024-12-02
> > >
> > > Thank you for responding so quickly to our comment. We do not take this for granted!
> > >
> > > > I strongly disagree with the authors' statement that "runtime is the accepted criterion for routing problems in the entire operations research community". The contrary is the case: the whole operations research community agrees that for most algorithms, a trade-off often exists between runtime and solution quality. If the authors screen papers published on VRPs, they will find two types: i) papers that improve on solution quality with runtime being a secondary objective. Many papers exist in top-tier journals like Transportation Science with worse runtimes but better solution quality; ii) papers that improve compared to existing papers upon runtime without or by only slightly worsening solution quality. It is unclear how one can support the authors' statement based on such observations.
> > >
> > > We strongly agree that with you that "for most algorithms, a trade-off often exists between runtime and solution quality". This is why we were so surprised that you implied that runtime is basically irrelevant (as long as it takes less than one night to solve an instance) in your initial review. Since there exists a trade-off between runtime and solution quality, limiting the runtime of all algorithms to the same value allows for a fair comparison of the approaches based only on the solution quality.
> > >
> > > > Even for human-in-the-loop problems, one might be willing to wait longer than a few seconds to interact with the algorithm. If not, the authors must present their work in such a context and show that swift interactions are beneficial in this case. I think the paper is far from even touching upon such a setting as acknowledged by the authors.
> > >
> > > For larger instances with 2000 customers, we give each approach 4 minutes per instance. This is much longer than just a few seconds. While there are many approaches in the ML community that work "instantly," NDS is not one of them.
> > >
> > > > I agree with the authors that "probably the algorithms have been tuned," but in a different setting under different boundary conditions and with different scope. Avoiding retuning in this case adds up more to the initially mentioned comparison of apples and oranges.
> > >
> > > It would be an apples-to-oranges comparison if we compared a tuned HGS with an untuned NDS. For the tuning to make sense, we would need to do it separately for each problem and problem size. This results in 12 scenarios. If we tune NDS, LKH, SISRs, and HGS, we end up with 48 tuning runs, which is not computationally feasible for us. Believe us, we would like nothing more than to tune these algorithms with SMAC or irace.
> > >
> > > > I am wondering why the authors are so defensive on their assumptions. One could have easily rerun some experiments and reported solution quality over different runtimes by simply saving the solutions after different time limits during one run, which would have allowed either for an even more substantial contribution statement if the proposed method finds new BKS or outperforms the existing methods after longer runtimes, or, alternatively, would have allowed for a more nuanced results discussion and contribution statement. Looking at the fact that the authors avoid such a comparison does not increase my trust in the claims made.
> > >
> > > The additional experiment you are asking here for makes perfect sense. We agree that anytime performance is interesting, and will provide visualizations of this in future versions. You did not ask for this experiment in your initial review though. You suggested that runtime is largely irrelevant in practice, noting that "in practice, one would not use such a time-based criterion as the problems studied are static problems usually solved in a day-ahead fashion, where solution times are not limited to seconds." Based on this, we assumed that an analysis of the anytime performance would not be of particular interest to you.

---

### Official Review · Reviewer_mZY4 · 2024-11-04

**Soundness:** 2
**Presentation:** 3
**Contribution:** 1
**Rating:** 3
**Confidence:** 5

**Summary:**

This paper proposes Neural Deconstruction Search (NDS), a novel method for vehicle routing problems that iteratively improves solutions by selectively removing and re-inserting nodes. A reinforcement learning-based policy network guides the removal process, while a greedy insertion strategy reconstructs efficient routes. NDS achieves competitive performance and improves solution quality compared to traditional approaches.

**Strengths:**

- Originality: The paper proposes a unique deconstruction-reconstruction framework for VRPs, leveraging a reinforcement learning policy to optimize solution quality in a novel way.
- Quality: The experiments are thorough, showing significant performance improvements across multiple VRP benchmarks and validating the approach’s robustness.
- Clarity: The paper is well-organized, with clear explanations and visuals that make the deconstruction and reconstruction process easy to understand.

**Weaknesses:**

- The paper introduces a deconstruction and re-insertion heuristic for VRP improvement, but it lacks a clear comparison with well-known heuristics, such as the 2-opt method, which also iteratively refines solutions. Providing an explanation of how the proposed approach differs from 2-opt would clarify the advantages of using a learning-based method for the deconstruction-recreation process.

- While the paper presents NDS as a novel learning-based improvement heuristic (Costa, 2020), it does not include comparisons with other learning-based approaches in the same category. A comparative analysis with similar methods would offer a more complete view of NDS’s performance and highlight its specific strengths and weaknesses within the context of learning-based VRP solvers.

- The current experiments are restricted to a specific subset of VRP problems, but VRP encompasses a wide variety of problem settings (see Berto, 2024). Testing the proposed approach on additional VRP problems, or explaining any limitations that prevent its application to other settings, would strengthen the generalizability of the method and provide clarity on its applicability across diverse VRP scenarios.

- The paper lacks details on how LEHD, BQ, and other learning-based solver baselines were trained and configured for this study. This is especially crucial because the reported performance for these methods differs from their original papers. Specifically, LEHD was originally trained on problems with up to 100 nodes; further clarification on how it was trained in this paper’s setting is necessary to interpret the experimental results accurately. Including these details would make the experimental setup more transparent and allow for better reproducibility and understanding of the baseline comparisons.

d O Costa, P. R., Rhuggenaath, J., Zhang, Y., & Akcay, A. (2020, September). Learning 2-opt heuristics for the traveling salesman problem via deep reinforcement learning. In Asian conference on machine learning (pp. 465-480). PMLR.

Berto, F., Hua, C., Zepeda, N. G., Hottung, A., Wouda, N., Lan, L., ... & Park, J. (2024). Routefinder: Towards foundation models for vehicle routing problems. arXiv preprint arXiv:2406.15007.

**Questions:**

Is the proposed deconstruction-reconstruction framework specifically designed to be effective only for VRP-type problems, or could it be adapted to other combinatorial optimization problems with different structural properties, such as Maximum Independent Set (MIS)? Have the authors considered or explored the potential adaptability of this approach to CO problems beyond VRP? Understanding any insights or limitations regarding its generalizability would clarify whether this framework could be broadly applicable across different types of optimization challenges.

---

> ### Author Response · Authors · 2024-11-22
> **Author Comment 1/2**
>
> Thank you for reviewing our paper and providing your feedback!
>
> We thank you for pointing out three critical strengths of the paper and we hope that you will adjust your score to reflect those strengths. We address the rest of your review below.
>
> > The paper introduces a deconstruction and re-insertion heuristic for VRP improvement, but it lacks a clear comparison with well-known heuristics, such as the 2-opt method, which also iteratively refines solutions.
>
> The LKH3 method is an improved 2-opt method (with some other tricks as well). HGS and SISRs are far superior even to LKH3 and all other simple heuristics for this problem (savings, etc.). There is no point to compare to them.
>
> We are curious why you believe that adding adding 2-opt as a baseline provides any additional information?
>
> > Providing an explanation of how the proposed approach differs from 2-opt would clarify the advantages of using a learning-based method for the deconstruction-recreation process.
>
> The destroy and repair paradigm we use and 2-opt are standard concepts known in the routing literature, hence we do not explain the differences, but it is possible we have not been clear enough somewhere. If you are having trouble understanding the NDS algorithm, we would appreciate knowing what parts are not clear so that we can improve these parts of the paper.
>
> > While the paper presents NDS as a novel learning-based improvement heuristic (Costa, 2020), it does not include comparisons with other learning-based approaches in the same category. A comparative analysis with similar methods would offer a more complete view of NDS’s performance and highlight its specific strengths and weaknesses within the context of learning-based VRP solvers.
>
> There are no learning-based improvement methods that come close or match the performance of HGS on the CVRP (and by proxy NDS). We hence focused on comparing to state-of-the-art OR methods only. If there any ML methods in particular that you have in mind, we can of course show them in the experimental results? We have not included extra ML algorithms to avoid unnecessary computation; we note that most papers in the field just copy/paste results from previous works and ignore the hardware/computation times, leading to awkward comparisons. Our goal is to have a clean comparison.
>
> > The current experiments are restricted to a specific subset of VRP problems, but VRP encompasses a wide variety of problem settings (see Berto, 2024). Testing the proposed approach on additional VRP problems, or explaining any limitations that prevent its application to other settings, would strengthen the generalizability of the method and provide clarity on its applicability across diverse VRP scenarios
>
> Having three applications has been a standard in the AI/ML literature for many years. We hope the reviewer will consider that it would be unfortunate for the community if papers where the "experiments are thorough" and show the "approach's robustness" are given reject scores based on criteria that are not widely accepted by the community. Let us also note that we have not cherry-picked VRP variants-- the three variants in our paper are the only ones we tried.
>
> We of course agree that results for a wider range of VRP variants would be interesting; more is always better. The work of Berto et al. (2024) specifically focuses on introducing a foundation model, so it is only natural that they would test on many problems. They use the RL4CO framework, which we are unable to use in this work.
>
> Our work clearly demonstrates that NDS can be used to solve various VRP variants by evaluating on three significantly different VRP variants. How many different VRP variants would we need to solve to convince you of significance of our method?
>
> > The paper lacks details on how LEHD, BQ, and other learning-based solver baselines were trained and configured for this study. This is especially crucial because the reported performance for these methods differs from their original papers. Specifically, LEHD was originally trained on problems with up to 100 nodes; further clarification on how it was trained in this paper’s setting is necessary to interpret the experimental results accurately. Including these details would make the experimental setup more transparent and allow for better reproducibility and understanding of the baseline comparisons.
>
> For LEHD and BQ, we use the model provided by the authors (both trained on instances with 100 nodes). We will update the paper and make this clear. To ensure a fair comparison, our test instances have been sampled from the same distribution as the instances used during training. At test time, we limit the search duration of LEHD by runtime (identical to NDS). For BQ we set the beam search size to 64. We believe that giving all methods a similar search budget allows for a more fair comparison.

---

> > ### Author Response · Authors · 2024-11-22
> > **Author Comment 2/2**
> >
> > > Is the proposed deconstruction-reconstruction framework specifically designed to be effective only for VRP-type problems, or could it be adapted to other combinatorial optimization problems with different structural properties, such as Maximum Independent Set (MIS)? Have the authors considered or explored the potential adaptability of this approach to CO problems beyond VRP? Understanding any insights or limitations regarding its generalizability would clarify whether this framework could be broadly applicable across different types of optimization challenges.
> >
> > This is an interesting question, and we plan to explore the use of our framework for a broader range of combinatorial optimization problems in the future. From a theoretical standpoint, it should be applicable, but assessing its true performance and determining whether it matches state-of-the-art methods will require extensive experimentation. We have focused this paper purely on routing problems to avoid making claims that we cannot yet support. We leave this for future work while noting that our current work already addresses a sufficiently diverse set of routing problems to justify publication in its present form.

---

### Meta-Review · Area_Chair_TwCx · 2024-12-19

**Metareview:**

This paper proposed a learning based deconstruction search policy for solving vehicle routing problems. It is based on a large neighborhood search framework, and the deconstruction policy is learned by deep reinforcement learning. The strengths of this paper are the simplicity of the proposed method and its good empirical performance. The major weaknesses of the paper are limited technical contribution (rely too much on the SISR framework), and insufficient evaluation. I agree that it is very important to demonstrate that the source of performance improvement is indeed from the learning part, instead of the SISR framework. I think the points raised by all reviewers are meaningful and relavent. Considering all reviews are negative and not changed after rebuttal, I recommend rejection.

**Additional Comments On Reviewer Discussion:**

Reviewers raised serious concerns regarding the technical novelty, experimental settings, and comparison to more relavent baselines. Authors provided detailed responses to each of the points, but unfortunately, none of the reviewers changed their mind. The final scores are consistently negative.

---

### Decision · Program_Chairs · 2025-01-22

Reject